# Evaluation of the Presence of Biofilms in Corrosive Points in Surgical Instruments after Reprocessing

**William Rosário [1], Taís Almeida [2], Bélgica Andrade [1], Idalina Aoki [3], Brunela Silva [3], Mariel Aramayo [3], Evandro Watanabe [4], Maíra Ribeiro [5], Camila Bruna [2,*] and Kazuko Graziano [2,*]**

1. Hospital Clínico San Borja Arriarán-Chile, Santiago 8360160, Chile
2. Nursing School of São Paulo University, São Paulo 05403-000, Brazil
3. Department of Chemical Engineering, Polytechnic School, University of São Paulo, São Paulo 05508-010, Brazil
4. School of Dentistry of Ribeirão Preto, São Paulo University, São Paulo 05508-000, Brazil
5. Clinical Hospital and Nursing School, Federal University of Minas Gerais, Belo Horizonte 31270-901, Brazil
* Correspondence: caquartim@yahoo.com.br (C.B.); kugrazia@usp.br (K.G.)

**Abstract:** Corrosive surgical instruments are routinely observed in central sterile services departments around the world. In addition to other risks, they can harbor microorganisms in the form of biofilms. Thus, this study aimed to evaluate whether biofilms intentionally formed at corrosion points on surgical instruments are removable by manual and automated cleaning followed by sterilization. Laboratory experiments were performed where samples of corroded surgical instruments in use in practice were evaluated for biofilm presence using a scanning electron microscope. No biofilms were observed in the samples subjected to manual and automated cleaning, nor in the samples in which there was no intentional biofilm formation. Residual organic matter without the presence of microorganisms was observed.

**Keywords:** surgical instruments; corrosion; biofilm; decontamination; sterilization

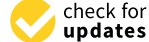



## 1. Introduction

Most surgical instruments are made of stainless steel, which meets the special conditions of use, cleaning, and sterilization. The most common types of steel are AISI-304 and AISI-420 [1]. They are made of several components, such as carbon, chromium, silicon, manganese, phosphorus, sulfur, and nickel, which are properly quantified in the formulation to be corrosion resistant when exposed to body fluids, cleaning solutions, sterilization/disinfection methods, and atmosphere [2,3].

Special electropolishing processes and passivation are used during the manufacturing of surgical instruments to reduce the probability of corrosion in these stainless-steel alloys throughout their usage [4]. Thus, in the context of the processing of healthcare products (PHP) performed by the central sterile services department (CSSD), finding instruments with the presence of corrosion indicates precarious conservation care, inadequate raw material, or exhaustion of the expected useful lifetime of the instruments.

The broadest definition of corrosion of surgical instruments is the deterioration of the instruments and their properties due to exposure to chemicals or electrochemical reactions between a material and its environment [5]. Corrosion can manifest as pitting corrosion (self-catalytic local breaking of the passive layer), crevice corrosion (frequent on screw backs), intergranular corrosion (failure in the microstructure of the metal), generalized corrosion (overall uniform removal of the passive layer by dissolution, resulting from exposure to very strong acid solutions), and stress corrosion (spreading of a crack as a result of the combination and synergistic interaction of mechanical stresses and corrosion reactions) [5–7].

The most common types of corrosion in surgical instruments are intergranular and pit corrosion. The first type is characterized by fast and localized development owing to a phenomenon called sensitization provoked by the precipitation of chromium carbide on the grain boundaries. When steel is exposed to high temperatures, chromium and carbon come together, forming chromium carbides in the intergranular region. These particles deplete the metal on chromium, reducing corrosion resistance [6]. They can also arise from welding or incorrect heat treatment [8]. The presence of chromium carbide particles can induce the nucleation of pits and their propagation.

Providing surgical instruments free of stains, oxidation spots, and corrosion should be part of the good practices of all CSSD. However, it is common to find surgical instruments that are in poor condition. Corrosion has been identified in 45.8% of surgical instruments in use, and the unregulated temperature of the autoclave and inadequate maintenance performed by unqualified people are causes of this occurrence [9].

In clinical practice, some surgeons reject instruments and surgical boxes that have corrosion on them. They even suspend the surgery with the justification that the damage to the instruments may compromise patient safety, cause iatrogenic events owing to the ease of formation and adhesion of biofilms, and result in the presence of endotoxins or release of heavy metals into the patient.

Of all the possible iatrogenic consequences that a corroded instrument may cause to the patient, there is no evidence that biofilms are associated with corrosion points, which may increase the severity of surgical site infection. Biofilms are a community of microorganisms surrounded by an amorphous extracellular material composed of exopolysaccharides of bacterial origin composed primarily of carbohydrates and proteins, but also with the presence of extracellular DNA and debris from dead cells [10]. Biofilms have been recognized as the greatest challenge to the PHP cleaning step because they progress to a stage of irreversible adhesion on the PHP surface and are only visible by methods, such as scanning electron microscopy (SEM) and confocal microscopy.

Thus, this study aimed to evaluate whether biofilms intentionally formed at corrosion points on surgical instruments are removable by manual and automated cleaning followed by sterilization.

## 2. Materials and Methods

Five instruments, three Mayo scissors, and one Halsted forceps, in use in a hospital and exhibiting corrosion in advanced stages, were selected as random samples for this study. To adapt the samples to SEM analysis, the instruments were fragmented at their corrosion points, identified with the naked eye, in sizes smaller than 1 cm. All fragments of samples from the experimental, positive, and negative control groups were previously cleaned and sterilized.

The preparation of the samples for analysis in the SEM was performed in aseptic technique using nitrile gloves and sterile tweezers. The samples were fixed on the microscope stubs using adhesive carbon tape. In a vacuum metallizer, a thin layer of gold was sputtered and deposited on each sample that was positioned in the SEM to obtain micrographs.

The samples of the experimental and positive control groups were subjected to intentional contamination by immersing them in a solution containing 3 mL of Sodium Thioglycolate, 4 mL of defibrinated sheep blood, 5 mL of 0.9% sodium chloride, and 1 mL microbial inoculum of *Pseudomonas aerugionosa* ATCC 27853 and *Enterococcus feccalis* ATCC 29212 ($1.5 \times 10^6$) to simulate a challenging scenario of microbial contamination and organic dirt, which is conducive for biofilm formation. Both are infection-causing, biofilm-forming microorganisms and were placed together as contaminants to simulate clinical practice, where both can be present in contaminations of surgical instruments.

The contaminant solution was distributed in sterilized test tubes with a volume of 12 mL each, and the five samples from the group considered experimental plus the three from the positive control group were individually immersed in each test tube. The tubes were then incubated for 6 h at a temperature of 35 °C to accelerate bacterial growth and

initiate biofilm formation. Subsequently, the samples remained exposed to the environment for 1 h, simulating a real situation in the CSSD, in which the material is not washed immediately. After the exposure, the samples of the experimental group were submitted to the cleaning process, according to the following standard operating procedures (SOPs), which simulate the sequence of steps recommended for cleaning surgical instruments:

1.   Pre-cleaning: A jet of water under pressure for 10 s;
2.   Ultrasonic washing (Sonica®, Padova, Italy) with enzymatic detergent containing five enzymes (Biozyme, Santiago, Chile) for 10 min at 40 °C;
3.   Rinsing with potable water and rubbing with a soft bristle brush on the surfaces with corrosion points of the specimens for 30 s;
4.   Clamped by pincers, the samples were deposited in a basket and subjected to the automated washing process in a pressure jet washer and thermal disinfector (Steelco, Treviso, Italy) with the following parameters: pre-cleaning for 2 min at 20 °C, cleaning for 5 min at 60 °C using an enzymatic detergent with five enzymes, rinsing for 1 min at 50 °C, thermal disinfection for 3 min at 90 °C, rinsing for 3 min at 20 °C, and drying for 30 min at 140 °C;
5.   The samples were individually wrapped in envelopes of surgical grade paper/film and were sterilized in an autoclave (Steelco, Italy) at 135 °C for 5 min; the cycles were monitored by type 5 integrator (3M®) and biological indicator (Attest 1292, 3M®), which showed satisfactory results of the cycle.

After rinsing and drying, the samples of positive control were packed in surgical-grade paper/film without going through the cleaning and sterilization steps. Three samples of instruments in use with advanced corrosion points that were not submitted to challenge contamination were submitted to the same processing protocol as the experimental samples characterizing the negative control group.

After processing, all samples were submitted to SEM. A total of 58 images for the experimental group, 22 for the positive control group, and 20 for the negative control, indicating the presence of biofilms, were registered. To confirm the authors' findings, three biofilm specialists independently analyzed the images.

## 3. Results

Figure 1, of one of the samples of the negative control group, shows magnified corrosion points of the surgical instruments after cleaning and sterilization, without images suggestive of biofilms.

During the compositional scan of one of the samples of the negative control group, an abnormally high concentration of carbon was found, and when the site was magnified by SEM, the image in Figure 2 was obtained. Because of the absence of images suggestive of microorganisms, it was not considered biofilm.

All positive control samples showed the presence of biofilms (Figure 3), demonstrating the success of the method for producing biofilms at corrosion points, as well as the possibility of biofilms forming at these points in the presence of organic matter and microorganisms.

No biofilms were visualized in any of the samples of the experimental control group, indicating success in removing biofilms by applying the proposed SOP (Figure 4).

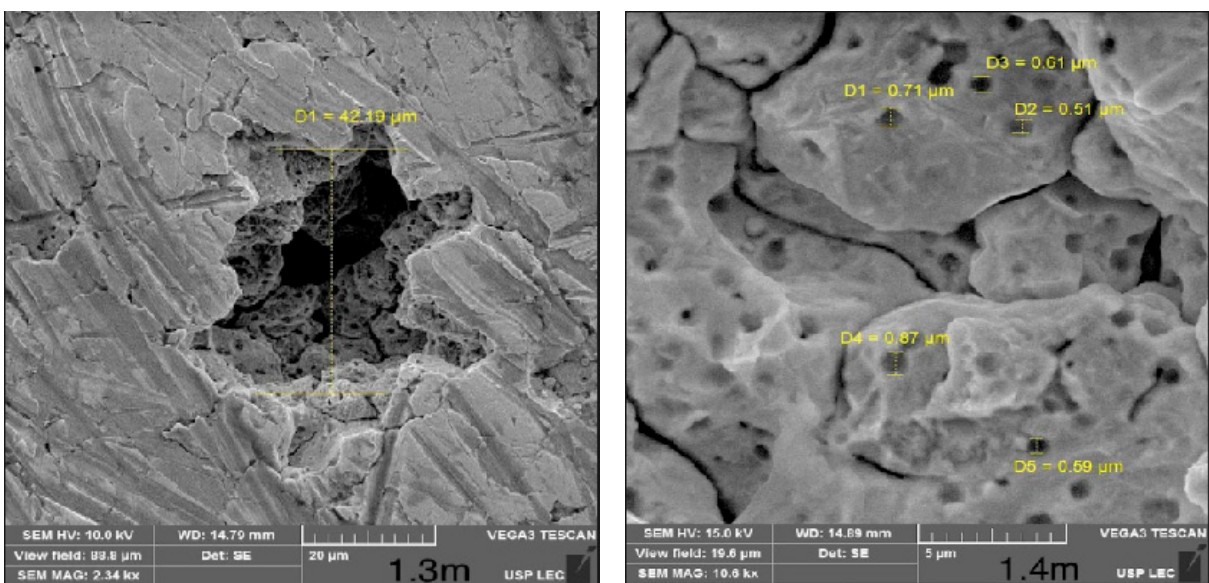

**Figure 1.** Magnified corrosion spot ("holes") of one of the negative control samples with measurements of the irregularities found, without images suggestive of biofilms ((**left**): 10.6 Kx magnification; scale bar: 5 μm; and (**right**): 2.34 Kx; magnification: 20 μm).

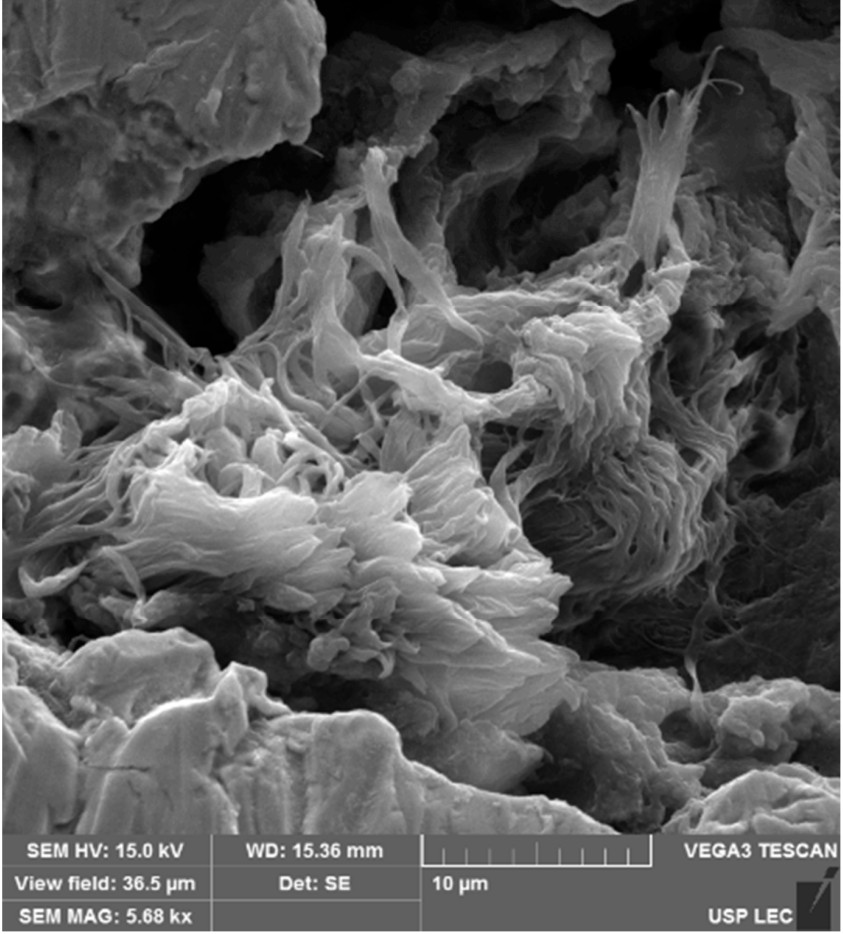

**Figure 2.** Magnified corrosion spot of one of the negative control samples with images suggestive of embedded dirt not recognized as biofilms (5.68 Kx magnification; scale bar: 10 μm).

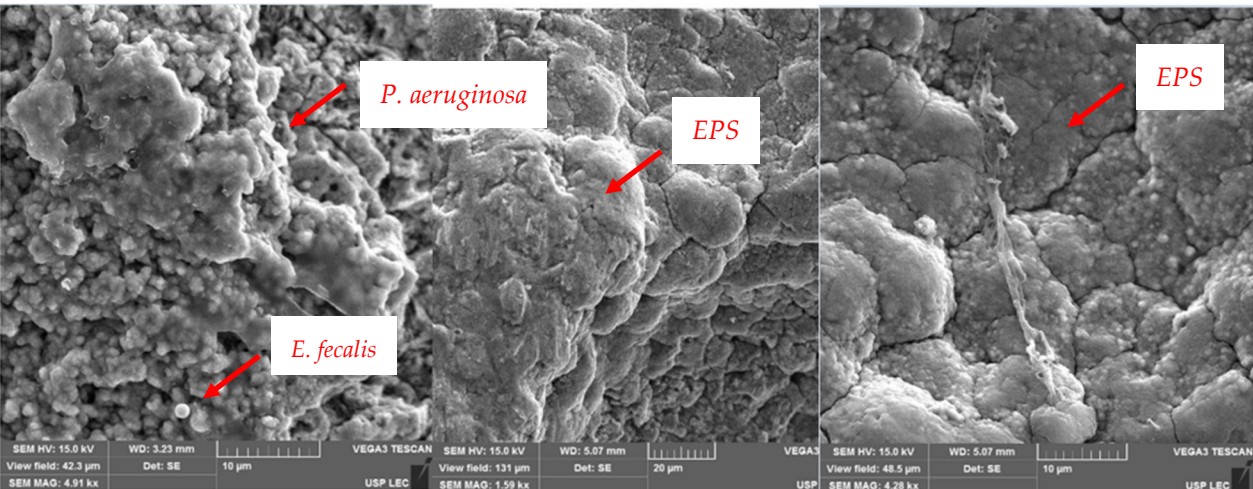

**Figure 3.** Biofilms visualized in the corrosion points of the instruments of the positive control group. The arrows point to microorganisms and extracellular polymeric substances (EPS). (**Left**): 4.91 Kx magnification; scale bar: 10 μm; (**Middle**): 1.59 Kx magnification; scale bar: 20 μm; (**Right**): 4.28 Kx magnification; scale bar: 10 μm.

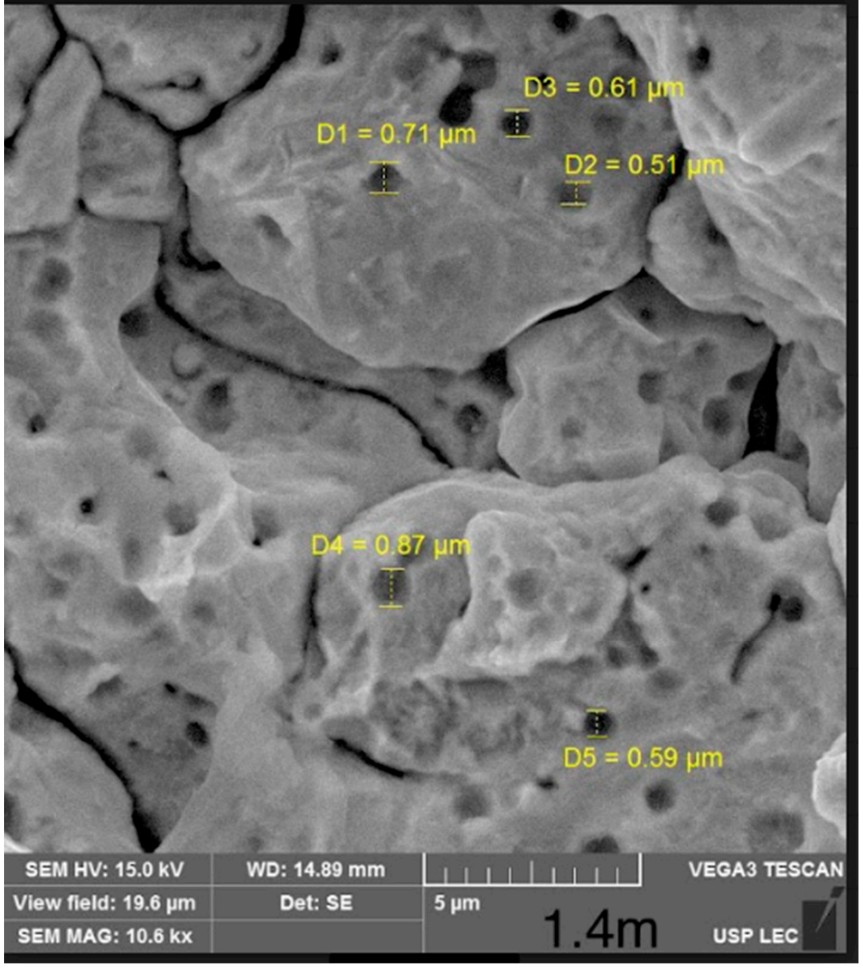

**Figure 4.** Magnified corrosion spot ("holes") of one of the samples of the experimental group without images suggestive of biofilms (10.6 Kx magnification; scale bar: 5 μm).

## 4. Discussion

Although there is a theoretical concern about the presence of microbial biofilms embedded in the corrosion points of surgical instruments, the present investigation refuted this evidence for the samples from the negative control and experimental groups. The negative control samples, which originated from corrosive instruments used in the practice, did not present biofilms, and those from the experimental group, although they had formed biofilms, inferring by the formation confirmed in the positive control group, also did not show signs of biofilms after manual and automated cleaning and sterilization procedures practiced by the CSSD. In one of the negative control samples, an adhered image was observed at the corrosion point (Figure 2), which was considered embedded dirt and not biofilm owing to the absence of microbial forms.

Considering the mechanisms of biofilm formation [10–12], biofilms may be inevitably formed on surgical instruments during operations and afterward, while awaiting cleaning in the CSSD. In this study, biofilm formed on corrosion areas was removed within 7 h.

Another research [13] also induced biofilm formation on complex surgical instruments and found organic matter after manual plus automated cleaning; however, no biofilms were found.

Another laboratory study [14] on the contamination of *Staphylococcus epidermidis* $10^6$ CFU/mL in surgical instruments demonstrated that the microbial load increased with time ($10^1$–$10^2$ CFU/cm$^2$, after 1 h; $10^4$ CFU/cm$^2$, after 12 h), showing the capacity for survival and replication of microorganisms while the instruments were soiled. The exopolysaccharide, characteristic of biofilms, was detected after 2 h and gradually increased thereafter. In contrast, the bacterial load was reduced by 1–2 log10 after manual cleaning and 1–3 log10 after automated cleaning, although the biofilms were not completely removed, unlike the results of the present study.

In this study, the instruments were submitted to a contamination cycle with consequent biofilm formation and a cleaning process, which may have influenced the effectiveness of the cleaning and detection of microorganisms. However, infectious outbreaks caused by the survival of microorganisms in the vegetative form in instruments sterilized by saturated steam under pressure at 134 °C cannot be ruled out. The plausible explanation for these outbreaks can only be attributed to biofilms and residual organic and inorganic matter that protected the microorganisms that would otherwise be eliminated at temperatures below 100 °C [15–17].

Although microorganisms were not detected in the present investigation, corrosion damage to the instruments allowed the accumulation of organic matter and carbon, which could represent a risk to the patients and favor the formation of biofilm afterward. These findings are not in accordance with the basic principle of PPS processing, which is to ensure that the surgical instruments used on a patient are in the same conditions as that of a new PPS in terms of functionality and absence of infectious risk and toxicity. Therefore, the results of this research reinforce that the cleaning of surgical instruments should be performed as soon as possible by the CSSD.

Moreover, other risks related to the corroded material must be considered, such as the risk of fracture in the intraoperative, the possibility of inadvertently retained material in the cavity, and threats to the operative technique. Risks related to the possible release of chemical elements that are part of the stainless-steel composition (carbon, chromium, silicon, manganese, phosphorus, sulfur, and nickel) and endotoxins, which could cause other adverse events to the surgical patients due to corrosion, not only of infectious nature but also cytotoxic and inflammatory, should be considered in future studies.

This study has the limitation of using two microbial species as contamination inoculum, and it is recognized that, in clinical practice, contamination occurs by multiple microorganisms. In practice, multiple uses of the instruments, associated with the delay in processing, can lead to a cyclic process of biofilm formation and hinder or prevent the effectiveness of the processing.

## 5. Conclusions

The biofilms intentionally formed at the corrosion points of the surgical instruments were removed by manual and automated cleaning performed within 7 h after contamination of the instruments; however, there were residues of organic matter at the corrosion points after these processes.

**Author Contributions:** Conceptualization: W.R., T.A., C.B. and K.G. Methodology: B.A., I.A., B.S., M.A., E.W., C.B. and K.G. Formal analysis: W.R., T.A., B.A., I.A., B.S., M.A., E.W., C.B., M.R. and K.G. Investigation: W.R., T.A., B.A., B.S., M.A., C.B. and K.G. Data curation: W.R., T.A., B.A., I.A., B.S., M.A., E.W., C.B., M.R. and K.G. Writing—original draft preparation: K.G.; Writing—review and editing: W.R., T.A., C.B., M.R. and K.G. Project administration: W.R., T.A. and K.G. All authors have read and agreed to the published version of the manuscript.

**Funding:** This research received no external funding.

**Institutional Review Board Statement:** Not applicable.

**Informed Consent Statement:** Patient consent waved due to no human being involved in the study.

**Data Availability Statement:** No data availability statement; data available on request.

**Acknowledgments:** Special thanks to Karen Vickery and Dayane de Melo Costa for their opinions on the presence of biofilms in scanning electron microscopic micrographs. The authors would like to thank the CNPq—National Council for Scientific and Technological Development—Brazil for the scholarships (grants # 140187/2017-0 and 310504/2020-1) and CAPES scholarship – Brazil [Coordination for the Improvement of Higher Education Personnel (grant number: 88887.507764/2020-00)].

**Conflicts of Interest:** The authors declare no conflict of interest.

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
