# Peer review of "Evaluation of the Presence of Biofilms in Corrosive Points in Surgical Instruments after Reprocessing"

_2673-947X, doi:10.3390/hygiene2040022_

Round 1

Reviewer 1 Report

This article tried to report the effect of cleaning on forming biofilms of surgical instruments. The authors mentioned “this study aimed to evaluate whether biofilms intentionally formed at corrosion points on surgical instruments are removable by manual and automated cleaning followed by sterilization”. However, this presentation is shortage in experimental results indicating the biofilms formed in the corroded sites, and there is no explanation about how the authors detected or analyzed the corroded sites of samples. I think you should present the evidences what the spot(s) formed biofilms were corresponded to the corroded position, additionally, you should analyze whole area including corroded sites and no damage (non-corroded) ones.

Based on my opinion, I assume this title is not reflected to this research. I recommend the authors should consider more suitable title.

Author Response

The corroded sites were identified on the samples before, with the naked eye, and then subjected to SEM, we have clarified this in the text. It was not the aim of the study to compare the areas with corrosion and the areas without corrosion in relation to the biofilm.

We hope that this clarification shows that the title does not need revision.

Reviewer 2 Report

Rosario et al. used SEM and reported the presence of biofilm in the corrosion points in surgical instruments and the biofilms could be removed by manual and automated cleaning procedures. Although this study is interesting and could be clinically important, I feel like more data might be needed to support their results.

1.       Why did the authors choose the two bacteria? Are the contamination relevant to the real clinical scenario? Were the two bacteria, P. aeruginosa and E. faecalis, inoculated together on the same instrument?

2.       Figure 3, can the authors find the two bacteria on the SEM images based on the morphology? P. aeruginosa would be rod shape while E. faecalis should like spheres.   

3.       All the data are based solely on SEM images. Could the authors plate the samples on TSA plate or blood agar plates to verify the presence of the bacteria? It would also be helpful to verify the effectiveness of cleaning procedures by the plate method. Plate method would also help verify whether the dust were bacteria in Figure 2.

4.       More information such as treatments, sample types could be added to figure captions to help readers understand the results. Please clearly indicate the scale bars.

5.       What are the holes in Figures 1 and 4?

Author Response

  1. Why did the authors choose the two bacteria? Are the contamination relevant to the real clinical scenario? Were the two bacteria, aeruginosa and E. faecalis, inoculated together on the same instrument?

One bacterium is gram-negative and another gram-positive. Both are tolerant to chemical and physical agents (difference in cell wall constituents and high capacity to form biofilm). Also, both bacteria are used together for instrument contamination, simulating a real contamination after the use of the instruments. E. faecalis and P. aeruginosa are related to environmental contamination (water, air conditioning, soap and disinfectants, places with humidity) and in some cases the hands of healthcare workers. We include this information in the text.

  1. Figure 3, can the authors find the two bacteria on the SEM images based on the morphology? aeruginosa would be rod shape while E. faecalis should like spheres.

In Figure 3, we indicate with arrows the bacteria.

  1. All the data are based solely on SEM images. Could the authors plate the samples on TSA plate or blood agar plates to verify the presence of the bacteria? It would also be helpful to verify the effectiveness of cleaning procedures by the plate method. Plate method would also help verify whether the dust were bacteria in Figure 2.

Each method of biofilm detection has advantages and disadvantages, and culture only provide indirect values. This was the reason we chose to perform SEM, since direct imaging of the biofilm provides information about its structural characteristics, its interaction with the surface, and spatial information. We could not perform live/dead SEM, and we did not culture the samples, but considered the presence of whole microorganisms as possible contaminants. Culture is not always able to isolate microorganisms, especially within biofilms.  

  1. More information such as treatments, sample types could be added to figure captions to help readers understand the results. Please clearly indicate the scale bars.

We add this information on the figures.

  1. What are the holes in Figures 1 and 4?

The "holes" are the corrosion points, we clarify in the images.

Round 2

Reviewer 1 Report

I hope this work will give precious information about biofilm infection diseases through medical devices to quit the wrong sterilizing procedure. 

Author Response

Thank you so much for your time in reviewing the document.

Reviewer 2 Report

The authors has largely addressed my concerns in this revised manuscript and I would recommend to publish this manuscript in Hygiene if there is no other issues. 

Author Response

Thank you so much for your time reviewing the document.